# Agreement between arterial and end-tidal carbon dioxide in adult patients admitted with serious traumatic brain injury

**Neil Sardesai**[1,2,3]*, **Owen Hibberd**[4], **James Price**[4], **Ari Ercole**[2,3], **Ed B. G. Barnard**[4,5]

**1** Emmanuel College, University of Cambridge, Cambridge, United Kingdom, **2** Division of Anaesthesia, Addenbrooke's Hospital, University of Cambridge, Cambridge, United Kingdom, **3** Cambridge Centre for Artificial Intelligence in Medicine, Cambridge, United Kingdom, **4** Emergency and Urgent Care Research in Cambridge (EUReCa), PACE Section, Department of Medicine, University of Cambridge, Cambridge, United Kingdom, **5** Academic Department of Military Emergency Medicine, Royal Centre for Defence Medicine (Research & Clinical Innovation), Birmingham, United Kingdom

\* nas60@cam.ac.uk

**Data Availability Statement:** Subject to the terms of the HRA approval and Cambridge University Hospitals NHS Foundation Trust policy, data cannot be shared by the authors. Researchers with

## Abstract

### Background

Low-normal levels of arterial carbon dioxide ($PaCO_2$) are recommended in the acute phase of traumatic brain injury (TBI) to optimize oxygen and $CO_2$ tension, and to maintain cerebral perfusion. End-tidal $CO_2$ ($ETCO_2$) may be used as a surrogate for $PaCO_2$ when arterial sampling is less readily available. $ETCO_2$ may not be an adequate proxy to guide ventilation and the effects on concomitant injury, time, and the impact of ventilatory strategies on the $PaCO_2$-$ETCO_2$ gradient are not well understood. The primary objective of this study was to describe the correlation and agreement between $PaCO_2$ and $ETCO_2$ in intubated adult trauma patients with TBI.

### Methods

This study was a retrospective analysis of prospectively-collected data of intubated adult major trauma patients with serious TBI, admitted to the East of England regional major trauma centre; 2015–2019. Linear regression and Welch's test were performed on each cohort to assess correlation between paired $PaCO_2$ and $ETCO_2$ at 24-hour epochs for 120 hours after admission. Bland-Altman plots were constructed at 24-hour epochs to assess the $PaCO_2$-$ETCO_2$ agreement.

### Results

695 patients were included, with 3812 paired $PaCO_2$ and $ETCO_2$ data points. The median $PaCO_2$-$ETCO_2$ gradient on admission was 0.8 [0.4–1.4] kPa, Bland Altman Bias of 0.96, upper (+2.93) and lower (-1.00), and correlation $R^2$ 0.149. The gradient was significantly greater in patients with TBI plus concomitant injury, compared to those with isolated TBI (0.9 [0.4–1.5] kPa vs. 0.7 [0.3–1.1] kPa, p<0.05). Across all groups the gradient reduced over time. Patients who died within 30 days had a larger gradient on admission compared to those who survived; 1.2 [0.7–1.9] kPa and 0.7 [0.3–1.2] kPa, *p*<0.005.

appropriate may apply for permission to access from CUH.dataresearch@nhs.net subject to HRA approval and Trust guidelines and review.

**Funding:** This research was supported by data analysts from the NIHR Cambridge Biomedical Centre (BRC 121520014).

**Competing interests:** The authors have declared that no competing interests exist.

## Conclusions

Amongst adult patients with TBI, the $PaCO_2$-$ETCO_2$ gradient was greater than previously reported values, particularly early in the patient journey, and when associated with concomitant chest injury. An increased $PaCO_2$-$ETCO_2$ gradient on admission was associated with increased mortality.

## Background

In the UK, 15% of major trauma patients require advanced airway and ventilatory management, where targeted neuroprotective measures are the cornerstone of reducing the impact of secondary brain injury [1]. The relationship between cerebral perfusion and arterial carbon dioxide ($PaCO_2$) is well understood. Unwarranted hypercapnia results in scerebral vasodilation and increases intracranial pressure, whilst hypocapnia can cause cerebral vasoconstriction and subsequent cerebral ischaemia [2]. Low-normal levels of $PaCO_2$ are recommended in the acute phase of TBI to optimize oxygen and $CO_2$ tension [2]. $PaCO_2$ values outside of a narrow therapeutic range are associated with increased mortality [3–7]. End-tidal $CO_2$ ($ETCO_2$) is often used as a surrogate for $PaCO_2$ when arterial sampling is less readily available in resource-constrained settings such as the prehospital environment [8–10]. The limited available data report low levels of agreement between $PaCO_2$ and $ETCO_2$ and in polytrauma and head-injured patients, suggesting that use of the $ETCO_2$ alone is not adequate to guide ventilation [9–14].

The $PaCO_2$–$ETCO_2$ gradient is substantially determined by the relative contribution of dead-space ventilation and is therefore influenced by ventilatory strategy [15]. The accepted gradient is considered to be 0.5kPa [16]. However, since the publication of ARDSNet in 2000, the gradient has not been reconsidered following lower tidal-volume ventilation strategies [17]. Reappraisals have consistently demonstrated heterogeneity in the agreement beween the $PaCO_2$ and $ETCO_2$, and variability in the gradient [18–23], which also appears to demonstrate prognostic value, particularly in the Intensive Care Unit (ICU) setting [23–25].

The effects on concomitant injury, time, and the impact of ventilatory strategies on the $PaCO_2$-$ETCO_2$ gradient are less well understood, and may guide future practice and realign current, potentially outdated, neuroprotective targets. The aim of this study was to describe the correlation and agreement between $PaCO_2$ and $ETCO_2$ in a large cohort of intubated major trauma patients with TBI.

## Methods

### Study design

This study was a retrospective analysis of prospectively-collected data of intubated adult major trauma patients with TBI, admitted to the East of England regional major trauma centre (MTC), 2015–2019. This study was undertaken at Cambridge University Hospitals NHS Foundation Trust (CUH), United Kingdom—the MTC for the East of England Trauma Network, covering a population of 6.3 million people over 20,000km$^2$.

### Inclusion criteria

Patients were included if they were 18 years or older, and sustained a serious or more severe TBI, defined as an Abbreviated Injury Scale (AIS) score $\geq 3$ for 'head', who underwent

endotracheal intubation either prehospital or in the emergency department, and were admitted to an ICU with ongoing mechanical ventilation.

## Exclusion criteria

$ETCO_2$ data without a paired $PaCO_2$ value within ten minutes were removed. Patients with $ETCO_2$ data without a $PaO_2$ or $FiO_2$ value (in order to calculate $PaO_2$-$FiO_2$ ratios) at the time of the reading were also excluded. Patients without any paired $PaCO_2$-$ETCO_2$ data points in the first 120-hours of admission to ICU were excluded.

## Data definitions

TBI was defined as an AIS score $\geq$3 for head. Isolated-TBI was defined as no AIS scores $\geq$1 in other anatomical body regions. TBI-plus was defined as an AIS $\geq$3 head together with at least one other anatomical body region AIS $\geq$1.

## Data collection

Demographics, mechanism of injury, injury severity (AIS and Injury Severity Score (ISS)), 30-day mortality, and functional outcome (Glasgow Outcome Scale (GOS)) data were obtained from the CUH Trauma Office records. Matched patient data were obtained from the CUH electronic medical record system (EMR, Epic Systems Corporation, Wisconsin, USA). Additional demographic characteristics, paired $PaCO_2$ and side-stream $ETCO_2$ was extracted from the patient EMR as close as possible to arrival in the ICU, and at 24, 48, 72, 96 and 120 hours after admission (or as near as possible to allow a close temporal link with blood gas data). $PaO_2$ and $FiO_2$ were also extracted concurrently with $PaCO_2$ to generate $PaO_2$/$FiO_2$ (P/ F) ratios, stratified into three bins to represent severity of Acute Respiratory Distress Syndrome (ARDS, mild: 26.6–40.0 kPa, moderate: 12.3–26.6 kPa, severe: <13.3 kPa) [26, 27]. Patient data was extracted on 27th August 2021. The authors did not have access to information that could identify individual participants during or after data collection.

## Primary outcome

The primary aim of this study was to report the correlation and agreement between $PaCO_2$ and $ETCO_2$ in intubated adult trauma patients with serious or more severe TBI on admission to the ICU.

## Secondary outcomes

The secondary outcomes were to: a) report the correlation and agreement of $PaCO_2$ and $ETCO_2$ values in isolated-TBI compared to patients with concomitant injuries (TBI+), b) to report temporal changes in this relationship over the first five days of admission, and c) to report the association between $PaCO_2$ and $ETCO_2$ gradient at ICU admission and 30-day mortality.

## Statistical analyses

Data manipulation and statistical analyses were performed using the Python programming language; significance was pre-defined at $p<0.05$, and no corrections were made for multiple comparisons. Complete case analysis was undertaken. Histograms were used to visually inspect the data for data quality and normality.

Basic demographic, mechanism of injury, and injury data are reported as number (percentage) and mean (+/- standard deviation) or median [interquartile range] as appropriate. $PaCO_2$

and $ETCO_2$ data are reported as mean (+/- standard deviation) and range. Comparisons of unpaired, normally distributed, continuous variables was undertaken with a two-tailed unpaired *t*-test (with Welch's correction), reported as a *t*-value and a *p*-value. Linear regressions have been performed to test for correlation and are reported as R-squared ($R^2$) with gradient of the slope (*m*).

Bland-Altman plots were used to test for agreement between paired $PaCO_2$ and $ETCO_2$ data and are reported as bias (95% confidence interval (95%CI)) with upper and lower limits of agreement. Bland-Altman plots were constructed for patients with TBI (AIS for head $\geq$3), as well as patients with TBI only (AIS for head$\geq$3, while other AIS scores = 0), and for patients with concomitant injuries (other AIS scores$>$0).

## Ethical review

Approval was obtained from the NHS Health Research Authority (HRA), protocol number—A095827. The study was locally registered with the CUH Safety and Quality Support Department.

## Results

### Demographics and injury

During the study period 1746 patients were eligible for inclusion. 1051 patients met predefined exclusion criteria and *n* = 695 patients were included in the final analysis; per protocol (Fig 1).

The median age was 48 [32–64] years old, and *n* = 523 (75.3%) were male. The median ISS was 29 [25–38], and 688 patients (99.0%) had an ISS >15. The most prevalent mechanism of injury was vehicle incident/collision (Table 1). Overall, 30-day mortality was 27.5% (191/695). The median interval between time of injury and ICU admission was 7.7 [5.3–10.5] hours.

### Primary outcome

The median $PaCO_2$ and $ETCO_2$ gradient in intubated adult trauma patients with serious or more severe TBI at ICU admission was 0.8 [0.4–1.4] kPa. Linear regression analysis demonstrated a correlation between $PaCO_2$ and $ETCO_2$ of $R^2$ = 0.149 (Fig 2). Blant Altman Bias comparing $PaCO_2$ and $ETCO_2$ was 0.96 (95% limits of agreement, -1.00 to +2.93) kPa (Fig 3).

### Secondary outcomes

**Injury burden comparison.** The $PaCO_2$ and $ETCO_2$ gradient was significantly greater in patients with a TBI plus concomitant injuries (TBI+) compared to those with isolated-TBI, 0.9 [0.4–1.5] kPa vs. 0.7 [0.3–1.1] kPa, p<0.05 (Figs 4 and 5).

**Temporal trends.** The mean $PaCO_2$-$ETCO_2$ gradient during the first 120-hours of admission reduced with time for all patient cohorts (Table 2). This reduction was most pronounced in patients with moderate-severe ARDS (Figs 6 and 7).

**Association with 30-day mortality.** The magnitude of the $PaCO_2$-$ETCO_2$ gradient was associated with increased mortality (Fig 5). Patients who died within 30 days had a larger gradient on admission compared to those who survived; 1.2 [0.7–1.9] kPa and 0.7 [0.3–1.2] kPa respectively, *p*<0.005.

## Discussion

This study demonstrates poor correlation between the $PaCO_2$ and $ETCO_2$ gradient for intubated adult patients with serious or more severe TBI. The presence of concomitant injuries was associated with a larger gradient. The magnitude of the gradient reduced during the first

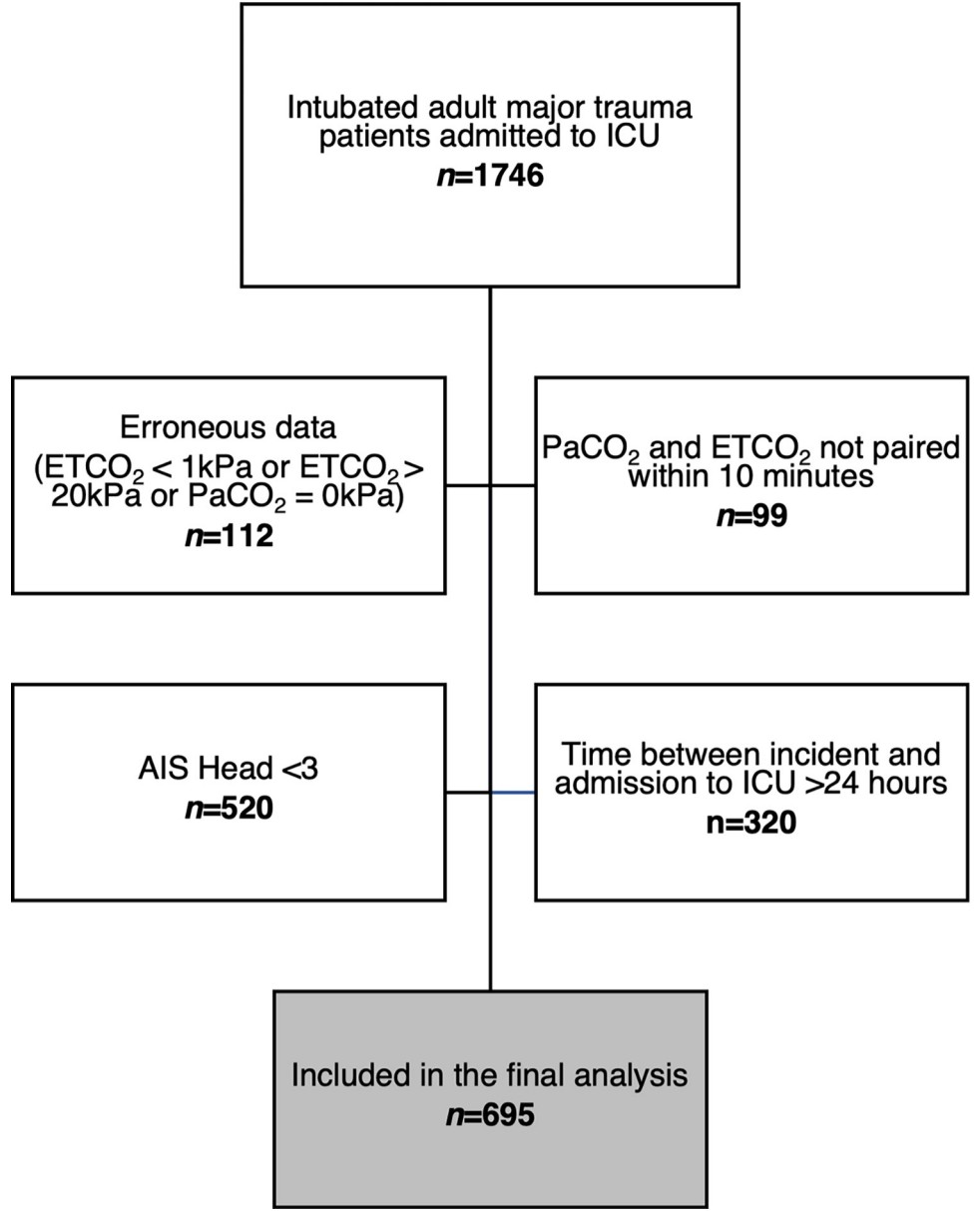

**Fig 1. Study flow diagram of intubated adult major trauma patients with serious or more severe TBI admitted to the intensive care unit at Cambridge University Hospitals NHS Foundation Trust; 2015–2019.** AIS—Abbreviated Injury Scale, ETCO2—End-tidal carbon dioxide, PaCO2—partial pressure of arterial carbon dioxide, ICU—Intensive Care Unit, TBI—Traumatic Brain Injury.

120 hours of admission, particularly in patients with moderate-severe lung injury. A larger $PaCO_2$ and $ETCO_2$ gradient on admission was associated with increased mortality.

Ventilatory optimisation and tight $PaCO_2$ control is paramount in the management of TBI. Subtle changes in $PaCO_2$ are closely associated with cerebral blood flow, and if poorly managed, with increased mortality [3–7]. Guidelines in resource limited environments without readily available access to arterial blood gas monitoring recommend the use of $ETCO_2$ as a surrogate to $PaCO_2$ to guide ventilation [16, 28]. Historically, the $PaCO_2$-$ETCO_2$ gradient has been defined as 0.5 kPa, a figure derived from healthy patients or in patients without traumatic

**Table 1. Demographics of intubated adult major trauma patients with serious or more severe TBI admitted to the Intensive Care Unit at Cambridge University Hospitals NHS Foundation Trust, 2015–2019; n = 695.**

| Gender | *n* (%) |
|---|---|
| Male | 523 (75.3%) |
| Female | 172 (24.7%) |
| **Age** | 48 [32–64] |
| **Mechanism of injury** | |
| **Motor vehicle collision** | 342 (49.2%) |
| **Fall** | 285 (41.0%) |
| **Blow(s) without weapon** | 59 (8.5%) |
| **Other** | 9 (1.3%) |
| **Injury Severity Score (ISS)** | |
| **ISS 9–15** | 7 (1.0%) |
| **ISS >15** | 688 (99.0%) |
| **30-day mortality** | |
| **Alive** | 504 (72.5%) |
| **Dead** | 191 (27.5%) |
| **Glasgow Outcome Score** | |
| **1 (Death)** | 196 (28.2%) |
| **2 (Prolonged Disorder of Consciousness)** | 1 (0.1%) |
| **3 (Severe Disability)** | 99 (14.2%) |
| **4 (Moderate Disability)** | 240 (34.5%) |
| **5 (Good Recovery)** | 159 (22.9%) |
| **Abbreviated Injury Score (AIS) for 'head'** | |
| **3 (Serious)** | 64 (9.2%) |
| **4 (Severe)** | 164 (23.6%) |
| **5 (Critical)** | 467 (67.2%) |

injury [29–33]. Despite this commonly-quoted figure cited in guidelines [16, 28], several recent studies have challenged the reliability of this surrogate marker [18–23]. Whilst this study demonstrates a median gradient of 0.8 [0.4–1.4] kPa, not dissimilar to previous reports, the Bland Altman analysis and correlation co-efficient ($R^2$ = 0.149) indicate that the use of $ETCO_2$ to guide ventilation early in this disease process is a blunt tool.

In addition to poor correlation in the overall cohort, the $PaCO_2$-$ETCO_2$ gradient was significantly greater in patients with TBI and concommitent injuries, compared to patients with isolated TBI, demonstrating the relationship between increased ventilation-perfusion (VQ) mismatch in patients with multi-system injury. Previous work has demonstrated that concommitent injury is associated with larger $PaCO_2$-$ETCO_2$ gradients particularly for patients with chest injuries due to the increased alveolar deadspace secondary to reduced lung perfusion [13, 15]. Whilst AIS scores are quoted in previous work to describe the extent of chest injury in patient cohorts [11], this retrospectively applied score is clearly not available to clinicians during the early phases of trauma resuscitation, and therefore other markers of lung injury are required to guide ventilatory adjustments. In this study, $PaO_2$/$FiO_2$ (P/F) ratios were used as a marker of lung injury and stratified into three bins to represent severity of ARDS (Mild 26.6–40.0 kPa, Moderate 12.3–26.6 kPa, Severe <13.3 kPa).(27)(28) Patients with more severe lung injury had a significantly larger $PaCO_2$-$ETCO_2$ gradient than those with less severe lung injuries and patients with isolated brain injury. The novelty in which this study categories lung injury makes it challenging to contrast these data with others' work. However, previous work that has utilised AIS scoring as a marker of lung injury support these findings [13]. The

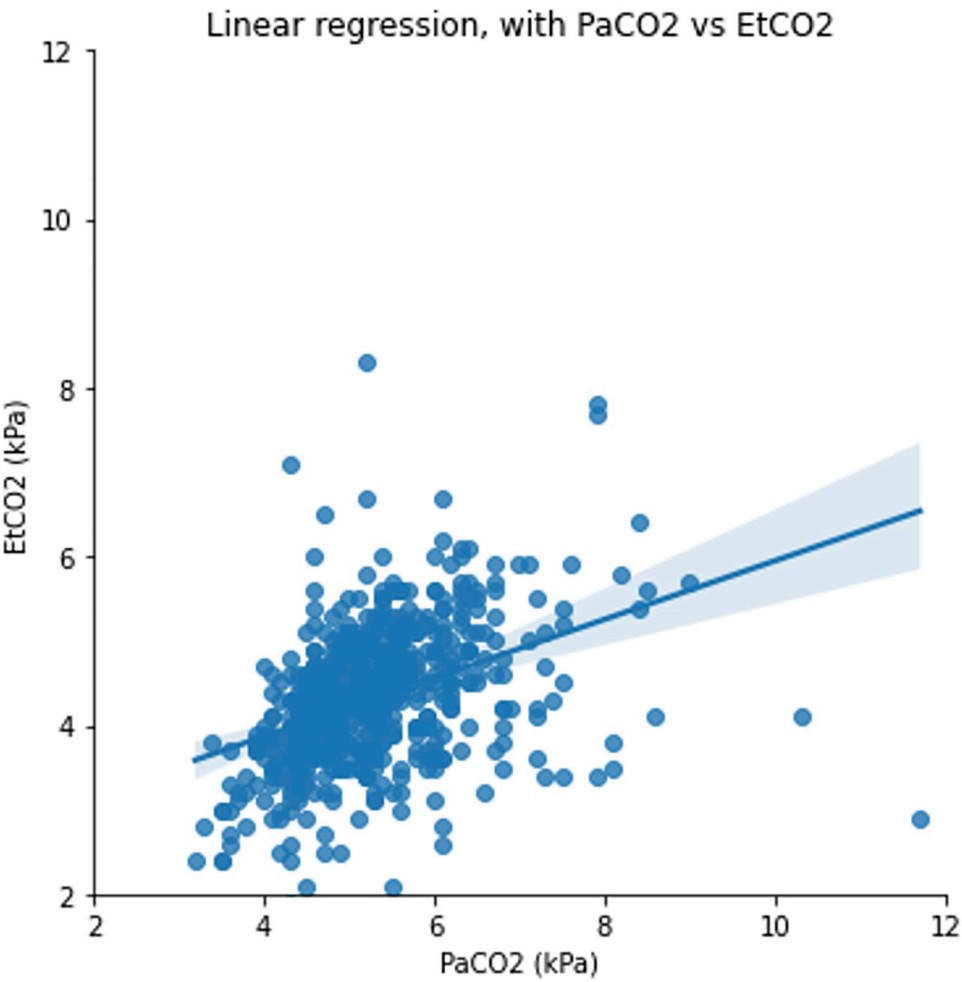

**Fig 2. Linear regression plot of PaCO2 and ETCO2 agreement in intubated adult major trauma patients with serious or more severe TBI admitted to the intensive care unit at Cambridge University Hospitals NHS Foundation Trust; 2015–2019.** ETCO2—End-tidal carbon dioxide, PaCO2 –partial pressure of arterial carbon dioxide, TBI—traumatic brain injury.

association between TBI and lung injury with mortality has been previously reported [34]. Including only a relatively small sample size, Robba *et al*. were able to demonstrate the P/F ratio to be a strong and independent risk factor for cerebral hypoxia and mortality in TBI patients [34]. These results imply the importance of managing cerebral perfusion and employing ventilation strategies appropriate for the P/F ratio and $PaCO_2$-$ETCO_2$ gradient.

The $PaCO_2$-$ETCO_2$ gradient in its earliest phase, the prehospital environment [14, 35–37], has been previously explored by the authors of this study [9, 10]. The correlation observed in the prehospital and emergency department phase ($R^2 = 0.23$, $p = 0.002$) prior to ICU admission demonstrates even worse agreement than observed in this study, indicating that with time, and the healing of tissue injury, it is antipated that the increase in alveolar perfusion and improvement in gas exchange, the $PaCO_2$-$ETCO_2$ gradient becomes smaller. This phenomenon has also been described in the paediatric population by Yang *et al*. where the gradient was larger in the first eight hours of admission compared with the subsequent nine to 24 hours of admission [18]. During the 120 hour observation period in this study, the gradient reduces, more significantly in patients with severe lung injury, however fails to reach a baseline gradient quoted in helathy volunteers of 0.5 kPa.

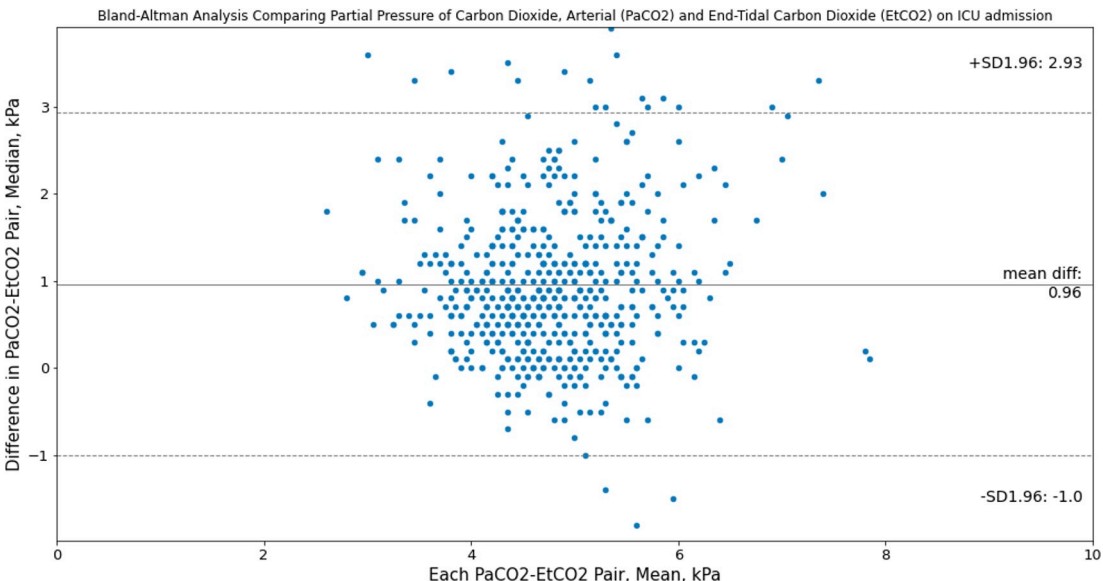

**Fig 3. Bland-Altman analysis comparing PaCO2 and ETCO2 agreement in intubated adult major trauma patients with serious or more severe TBI admitted to the intensive care unit at Cambridge University Hospitals NHS Foundation Trust; 2015–2019.** AIS—Abbreviated Injury Scale, ETCO2—End-tidal carbon dioxide, PaCO2—partial pressure of arterial carbon dioxide, ICU—Intensive Care Unit, TBI—Traumatic Brain Injury.

Recent work has explored the association between the $PaCO_2$-$ETCO_2$ gradient and mortality [19]. Within the first 24 hours after admission, the $PaCO_2$-$ETCO_2$ gradient has been demonstrated to be independently associated with increased mortality, with a three times increase in mortality for each increase of 1 kPa in the gradient [19]. Similarly, in this study, a larger $PaCO_2$-$ETCO_2$ gradient on admission was associated with increased mortality, for patients with isolated TBI and also those with concomittent injuries. This may also suggest a relationship between the severity of the head trauma and the $PaCO_2$-$ETCO_2$ gradient, however the severity of head injury in this cohort was not evaluated. These findings support the emerging evidence, and although cut-points have yet to be consistently identified, these preliminary

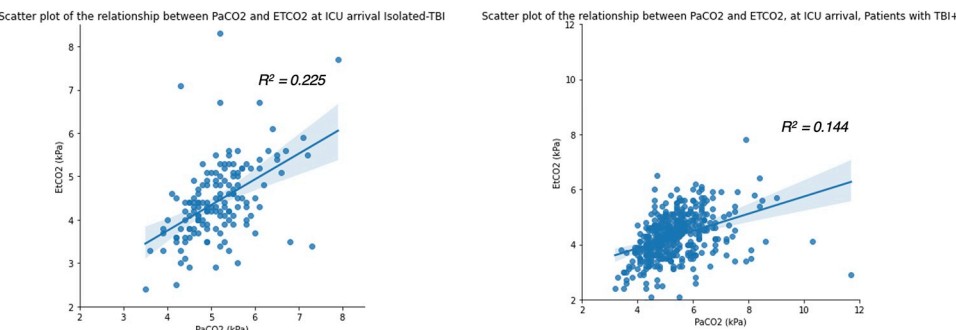

**Fig 4. Linear regression plots illustrating PaCO2 and ETCO2 agreement in patients with Isolated-TBI versus TBI +, admitted to the Intensive Care Unit at Cambridge University Hospitals NHS Foundation Trust; 2015–2019.** ETCO2—End-tidal carbon dioxide, PaCO2 –partial pressure of arterial carbon dioxide, TBI—traumatic brain injury, Isolated-TBI—patients with serious or more severe isolated TBI, with no concomitant injuries (AIS ≥3 'head' only), TBI+—patients with serious or more severe TBI plus concomitant injuries (AIS ≥3 'head' together with at least one other anatomical body region AIS ≥1).

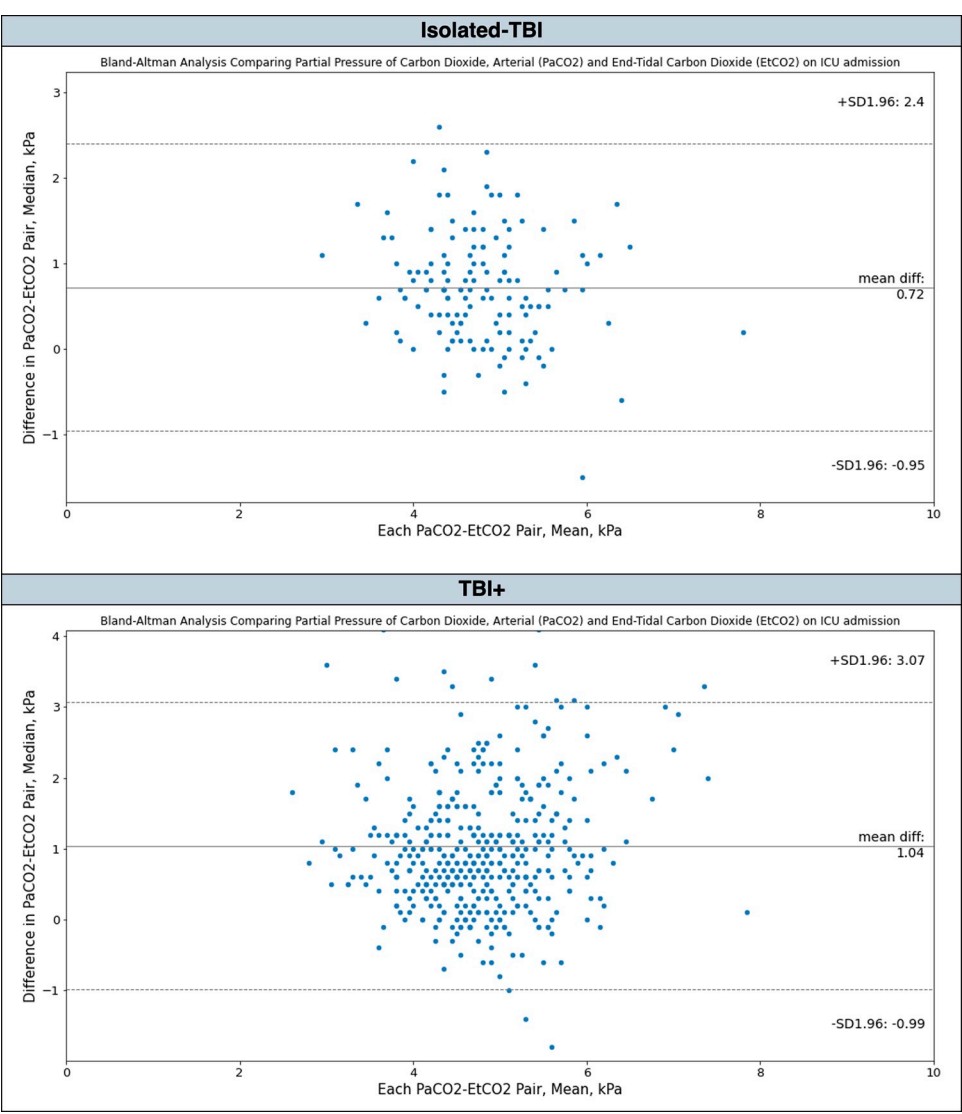

**Fig 5. Bland-Altman analysis illustrating PaCO2 and ETCO2 agreement in patients with Isolated-TBI versus TBI +, admitted to the Intensive Care Unit at Cambridge University Hospitals NHS Foundation Trust; 2015–2019.**
ETCO2—End-tidal carbon dioxide, PaCO2 –partial pressure of arterial carbon dioxide, TBI—traumatic brain injury, Isolated-TBI—patients with serious or more severe isolated TBI, with no concomitant injuries (AIS ≥3 'head' only), TBI+—patients with serious or more severe TBI plus concomitant injuries (AIS ≥3 'head' together with at least one other anatomical body region AIS ≥1).

**Table 2. Comparison of temporal trends in the PaCO2-ETCO2 gradient, amongst adult trauma patients with TBI, Isolated-TBI, and TBI+, admitted to the Intensive Care Unit at Cambridge University Hospitals NHS Foundation Trust; 2015–2019.**

| TBI | | | |
|---|---|---|---|
| | **PaCO2 (kPa)** | **ETCO2 (kPa)** | **Gradient (kPa)** | **Percentage of gradients > 1 kPa (%)** |
| **0 hours** | 5.2 [4.7–5.7] | 4.3 [3.8–4.8] | 0.8 [0.4–1.4] | 38.0 |
| **24 hours** | 5.1 [4.6–5.6] | 4.2 [3.7–4.8] | 0.7 [0.4–1.2] | 33.9 |
| **48 hours** | 5.2 [4.8–5.6] | 4.4 [3.9–4.9] | 0.7 [0.3–1.3] | 33.6 |
| **72 hours** | 5.3 [4.8–5.8] | 4.5 [4.0–5.0] | 0.8 [0.4–1.3] | 32.5 |
| **96 hours** | 5.3 [4.9–5.9] | 4.5 [4.0–5.2] | 0.8 [0.4–1.2] | 31.8 |
| **120 hours** | 5.3 [4.8–5.8] | 4.6 [4.0–5.2] | 0.7 [0.3–1.2] | 31.8 |

*(Continued)*

**Table 2.** (Continued)

| Isolated-TBI | | | |
|---|---|---|---|
| | **PaCO2 (kPa)** | **ETCO2 (kPa)** | **Gradient (kPa)** | **Percentage of gradients > 1kPa (%)** |
| **0 hours** | 5.1 [4.7–5.5] | 4.3 [3.9–4.9] | 0.7 [0.3–1.1] | 29.5 |
| **24 hours** | 5.0 [4.6–5.5] | 4.3 [3.8–4.8] | 0.7 [0.3–1.0] | 20.8 |
| **48 hours** | 5.1 [4.6–5.5] | 4.3 [3.8–4.9] | 0.7 [0.3–1.1] | 26.3 |
| **72 hours** | 5.2 [4.8–5.5] | 4.5 [3.8–5.0] | 0.7 [0.3–1.1] | 27.1 |
| **96 hours** | 5.1 [4.7–5.4] | 4.3 [3.8–4.8] | 0.6 [0.3–1.1] | 26.0 |
| **120 hours** | 5.0 [4.5–5.4] | 4.2 [3.7–4.9] | 0.5 [0.2–1.1] | 30.6 |
| TBI+ | | | |
| | **PaCO2 [median] (kPa)** | **ETCO2 [median] (kPa)** | **Gradient [median] (kPa)** | **Percentage of gradients > 1kPa (%)** |
| **0 hours** | 5.2 [4.7–5.8] | 4.3 [3.7–4.8] | 0.9 [0.4–1.5] | 40.7 |
| **24 hours** | 5.1 [4.6–5.6] | 4.2 [3.7–4.7] | 0.8 [0.4–1.3] | 37.8 |
| **48 hours** | 5.2 [4.8–5.7] | 4.4 [3.9–5.0] | 0.8 [0.4–1.3] | 36.0 |
| **72 hours** | 5.3 [4.9–5.8] | 4.5 [4.0–5.0] | 0.8 [0.4–1.3] | 34.0 |
| **96 hours** | 5.4 [4.9–6.0] | 4.5 [4.1–5.2] | 0.8 [0.4–1.2] | 33.5 |
| **120 hours** | 5.4 [4.9–5.8] | 4.7 [4.2–5.2] | 0.7 [0.3–1.2] | 32.2 |

ETCO2—End-tidal carbon dioxide, PaCO2 –partial pressure of arterial carbon dioxide, TBI—traumatic brain injury, Isolated-TBI—patients with serious or more severe isolated TBI, with no concomitant injuries (AIS ≥3 'head' only), TBI+—patients with serious or more severe TBI plus concomitant injuries (AIS ≥3 'head' together with at least one other anatomical body region AIS ≥1).

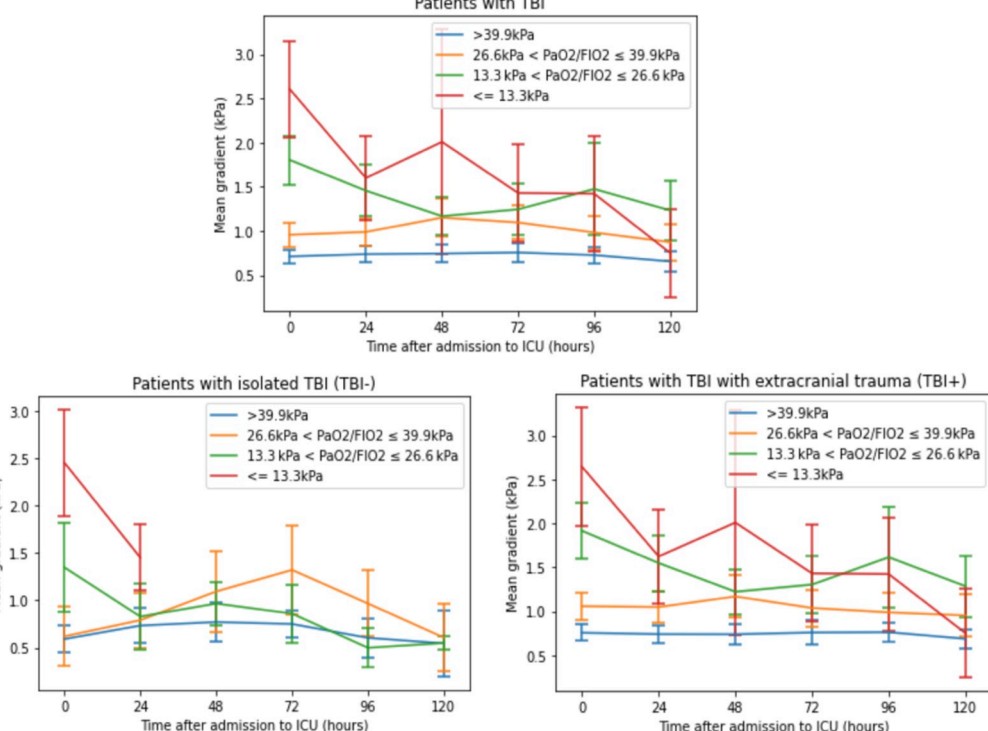

**Fig 6. Comparison of temporal trends in the PaCO2-ETCO2 gradient, stratified by the P/F ratio, amongst adult trauma patients with TBI, Isolated-TBI, and TBI+, admitted to the Intensive Care Unit at Cambridge University Hospitals NHS Foundation Trust; 2015–2019.** ETCO2- End-tidal carbon dioxide, PaCO2 –partial pressure of arterial carbon dioxide, TBI—traumatic brain injury, Isolated-TBI—patients with serious or more severe isolated TBI, with no concomitant injuries (AIS ≥3 'head' only), TBI+—patients with serious or more severe TBI plus concomitant injuries (AIS ≥3 'head' together with at least one other anatomical body region AIS ≥1).

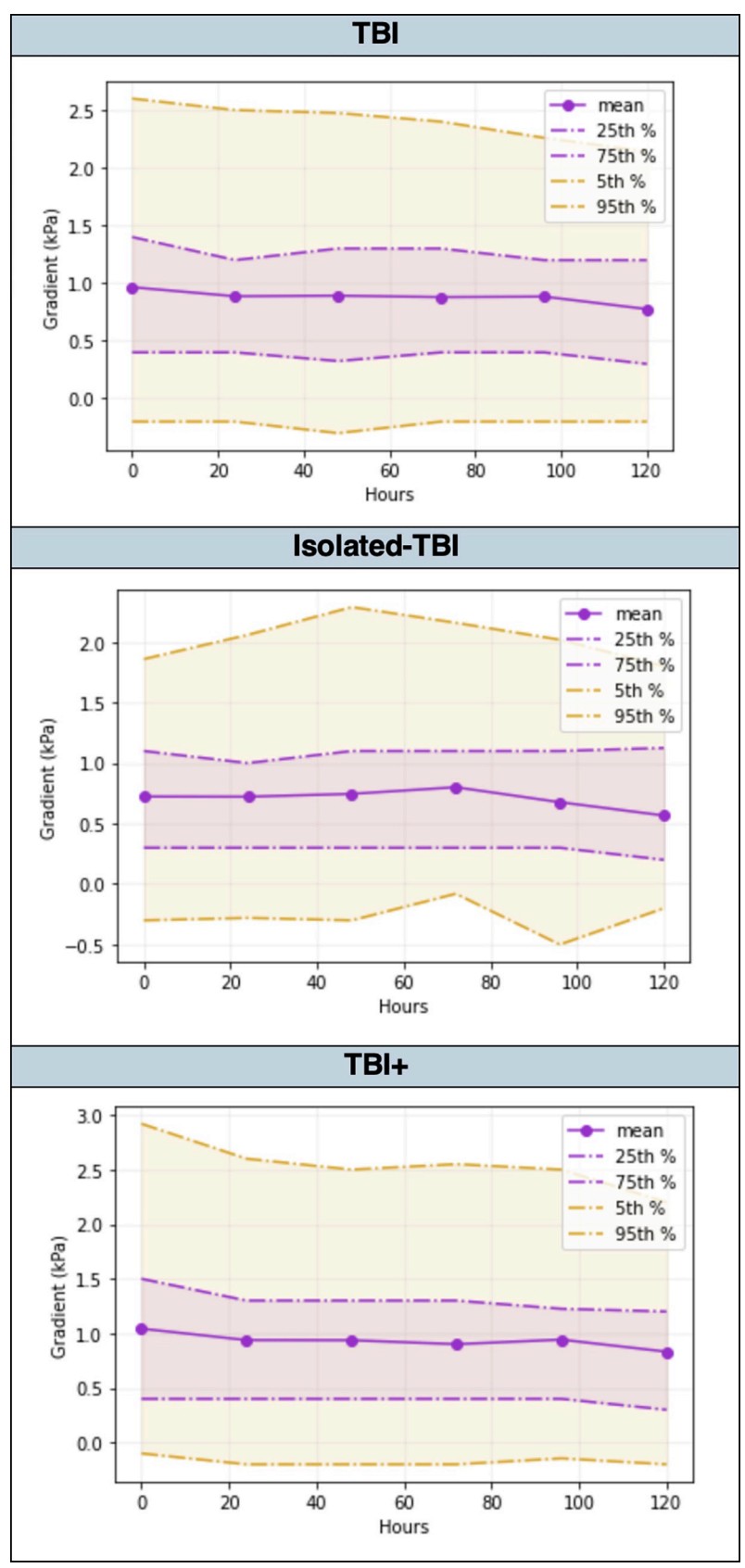

**Fig 7. Comparison of temporal trends in the PaCO2-ETCO2 gradient, stratified by quantile, amongst adult trauma patients with TBI, Isolated-TBI, and TBI+, admitted to the Intensive Care Unit at Cambridge University Hospitals NHS Foundation Trust; 2015–2019.** ETCO2- End-tidal carbon dioxide, PaCO2 –partial pressure of arterial carbon dioxide, TBI—traumatic brain injury, Isolated-TBI—patients with serious or more severe isolated TBI, with no concomitant injuries (AIS ≥3 'head' only), TBI+—patients with serious or more severe TBI plus concomitant injuries (AIS ≥3 'head' together with at least one other anatomical body region AIS ≥1).

results may be of use to system planners anticipating length of stay and prognostication when managing expectations.

Whist these data must be intereprerted as association and not causation, the importance of measuring the $PaCO_2$ early and adapting ventilation accordingly is clear. Despite this study being performed in a single trauma and neurosciences ICU, the large number of included patients allow for increased confidence in the results. Moreover, patients were only included on admission to ICU, therefore a survivor bias is likely to exist. However, patients that did not survive to ICU admission are likely to have catastrophic injuries in which the utility of the $PaCO_2$-$ETCO_2$ gradient is less clinically relevant.

## Conclusion

Amongst adult patients with TBI, the PaCO2-ETCO2 gradient was greater than previously reported values, particularly early in the patient journey, and when associated with concomitant injury. An increased PaCO2-ETCO2 gradient was associated with increased mortality.

## Acknowledgments

The authors would like to thank Junli Awit and Jakub Jaworski for their help in creating the EMR data extract. We would also like to acknowledge the assistance of Assiah Mahmood and Jacques Bowman of the CUH Trauma Office in compiling the original data.

## Author Contributions

**Conceptualization:** Neil Sardesai, Owen Hibberd, James Price, Ari Ercole, Ed B. G. Barnard.

**Data curation:** Neil Sardesai, Ari Ercole, Ed B. G. Barnard.

**Formal analysis:** Neil Sardesai, Owen Hibberd, James Price, Ari Ercole, Ed B. G. Barnard.

**Investigation:** Neil Sardesai, Owen Hibberd, James Price, Ari Ercole, Ed B. G. Barnard.

**Methodology:** Neil Sardesai, Owen Hibberd, James Price, Ari Ercole, Ed B. G. Barnard.

**Supervision:** Ari Ercole, Ed B. G. Barnard.

**Writing – original draft:** Neil Sardesai, Owen Hibberd, James Price, Ari Ercole, Ed B. G. Barnard.

**Writing – review & editing:** Neil Sardesai, Owen Hibberd, James Price, Ari Ercole, Ed B. G. Barnard.

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
