## [Decision Letter · Decision Letter 0]

24 Oct 2023

PONE-D-23-28025Agreement between arterial and end-tidal carbon dioxide in adult patients admitted with serious traumatic brain injuryPLOS ONE

Dear Dr. Sardesai,

Thank you for submitting your manuscript to PLOS ONE. After careful consideration, we feel that it has merit but does not fully meet PLOS ONE’s publication criteria as it currently stands. Therefore, we invite you to submit a revised version of the manuscript that addresses the points raised during the review process.

We look forward to receiving your revised manuscript.

Kind regards,

Wan-Jie Gu

Academic Editor

PLOS ONE

The authors would like to thank Junli Awit and Jakub Jaworski for their help in creating the EMR data extract. We would also like to acknowledge the assistance of Assiah Mahmood and Jacques Bowman of the CUH Trauma Office in compiling the original data. This research was supported by the NIHR Cambridge Biomedical Centre (BRC 121520014).

The authors received no specific funding for this work.

Reviewers' comments:

Reviewer's Responses to Questions

**Comments to the Author**

1. Is the manuscript technically sound, and do the data support the conclusions?

Reviewer #1: Partly

Reviewer #2: Yes

Reviewer #3: Yes

2. Has the statistical analysis been performed appropriately and rigorously? 

Reviewer #1: No

Reviewer #2: Yes

Reviewer #3: No

3. Have the authors made all data underlying the findings in their manuscript fully available?

Reviewer #1: No

Reviewer #2: Yes

Reviewer #3: Yes

4. Is the manuscript presented in an intelligible fashion and written in standard English?

Reviewer #1: Yes

Reviewer #2: Yes

Reviewer #3: Yes

5. Review Comments to the Author

Reviewer #1: Thank you for the opportunity to review this very hard work with valuable data. The authors analyzed the PaCO2-EtCO2 gradient in patients with TBI, and it would have been helpful if the significance of examining the PaCO2-EtCO2 gradient in patients with TBI could have been more clearly described, such as the rationale for assuming that the PaCO2-EtCO2 gradient is different in patients with TBI than in other patients, and the clinical significance of the PaCO2-EtCO2 gradient in TBI.

1. Abstract: The word "gradient" in the Results section should clearly state what gradient it is.

2. Abstract: “The gradient was significantly greater in patients with TBI plus concomitant injury, compared to those with isolated TBI”: Data should be presented.

3. Abstract: “Patients with TBI who died had a significantly larger gradient compared to those who survived”: Data should be presented.

4. Background: Do you hypothesize that the gradient between PaCO2 and EtCO2 is different between TBI and other patient groups? I would like to know the rationale for why it is necessary to examine the PaCO2-EtCO2 gradient in TBI patients.

5. Methods: Are there no data on whether there was spontaneous respiration or mode of ventilation at the time of data collection? The presence or absence of spontaneous breathing and mode of ventilation can have a significant effect on the PaCO2-EtCO2 gradient. In particular, EtCO2 values are highly variable when spontaneous breathing is present.

6. Results: The authors stated that “The PaCO2 and ETCO2 gradient was significantly greater in patients with a TBI plus concomitant injuries (TBI+) compared to those with isolated-TBI”. However, the difference is very small and the reviewer does not consider this difference to be clinically meaningful (0.9 [0.4-1.5] kPa vs. 0.7 [0.3-1.1] kPa.

7. Results; Temporal trends: At what point in time the P/F ratio was used to classify the severity of ARDS?

8. Results: Association with 30-day mortality: Does Figure 5 show a relationship between gradient and mortality? Please confirm. Given that the PaCO2-EtCO2 gradient is greater in patients with more severe head trauma, it is not surprising that the PaCO2-EtCO2 gradient is greater in patients who died compared to those who survived, which suggests that we are not looking at the relationship between the PaCO2-EtCO2 gradient and mortality, but rather only at the relationship between the severity of head trauma and the PaCO2-EtCO2 gradient.

9. Conclusion: Chest injury is not mentioned in either the Methods or the Results, and if you are going to include the following in the Conclusion, you should explain how you analyzed the association between the presence of chest injury and PaCO2-EtCO2: “Amongst adult patients with TBI, the PaCO2-ETCO2 gradient was greater …when associated with concomitant chest injury.”

Reviewer #2: Thank you very much for the opportunity to review this paper. I read it with great interest.

The relationship between etCO2 and PaCO2 is an essential and sometimes highly underestimated topic in the initial management of TBI; it is of great interest to clinicians (EMS and ICU).

I have no major concerns about the paper; a few minor points that the paper could benefit from:

p4: order of references not correct; 13/14 before 10...

p4, line 62, a very recent study on this topic could be added (Knapp, J., Doppmann, P., Huber, M. et al. Pre-hospital endotracheal intubation in severe traumatic brain injury: ventilation targets and mortality-a retrospective analysis of 308 patients. Scand J Trauma Resusc Emerg Med 31, 46 (2023). https://doi.org/10.1186/s13049-023-01115-8)

p5 and Results: a collection of AIS groups by organ system would allow a more detailed analysis (adjustment of results). Pre-hospital times? On-scene Time intervals? Concerning the etCO2-PaCO2 gap, haemodynamics plays a central role! Haemodynamics or Lac would, therefore, be significant covariates for a precise analysis. Please go into this in more detail or add it to the limitations. Another reason for an increased gap could be (too) high tidal volumes; are ventilation parameters available here? (If not, please discuss this and/or include it in the limitations).

p14. Please describe "concomitant injuries" in more detail here; primarily, thoracic injuries are to be mentioned and not, for example, extremity injuries.

Reviewer #3: The manuscript reported that amongst adult patients with TBI, the PaCO2-ETCO2 gradient was greater than previously reported values, particularly early in the patient journey, and when associated with concomitant chest injury. An increased PaCO2-ETCO2 gradient on admission was associated with increased mortality. These provided useful information for the clinical treatment of TBI patients in ICU. Major suggestions include:

1. The study showed that PaCO2-ETCO2 gradient in TBI + subgroup is lager than that of isolated -TBI subgroup. Please explain the difference in DISCUSSION

2. The magnitude of the PaCO2-ETCO2 gradient was associated with increased mortality (Figure 5). Patients who died within 30 days had a larger gradient on admission compared to those who survived; 1.2 [0.7-1.9] kPa and 0.7 [0.3-1.2] kPa respectively, p<0.005. Please explain it and provide useful suggestions for this kind of patients (e.g., ventilation management)

6. PLOS authors have the option to publish the peer review history of their article (what does this mean?). If published, this will include your full peer review and any attached files.

Reviewer #1: No

Reviewer #2: No

Reviewer #3: No

---

## [Author Response · Author response to Decision Letter 0]

13 Nov 2023

Thank you for your comments and review.

In terms of the funding, the research was supported by data analysts from the NIHR Cambridge Biomedical Centre. The data analysts retrieved the data for no cost. As instructed, we have removed the funding section of the manuscript. Please could you include the following statement in the online submission form: “This research was supported by data analysts from the NIHR Cambridge Biomedical Centre (BRC 121520014).”

Please see the 'Response to Reviewers' letter for additional comments. Many thanks.

---

## [Decision Letter · Decision Letter 1]

28 Dec 2023

Agreement between arterial and end-tidal carbon dioxide in adult patients admitted with serious traumatic brain injury

PONE-D-23-28025R1

Dear Dr. Sardesai,

We’re pleased to inform you that your manuscript has been judged scientifically suitable for publication and will be formally accepted for publication once it meets all outstanding technical requirements.

Kind regards,

Wan-Jie Gu

Academic Editor

PLOS ONE

Additional Editor Comments (optional):

Reviewers' comments:

Reviewer's Responses to Questions

**Comments to the Author**

1. If the authors have adequately addressed your comments raised in a previous round of review and you feel that this manuscript is now acceptable for publication, you may indicate that here to bypass the “Comments to the Author” section, enter your conflict of interest statement in the “Confidential to Editor” section, and submit your "Accept" recommendation.

Reviewer #1: All comments have been addressed

Reviewer #2: All comments have been addressed

Reviewer #3: All comments have been addressed

2. Is the manuscript technically sound, and do the data support the conclusions?

Reviewer #1: Yes

Reviewer #2: Yes

Reviewer #3: Yes

3. Has the statistical analysis been performed appropriately and rigorously? 

Reviewer #1: Yes

Reviewer #2: Yes

Reviewer #3: Yes

4. Have the authors made all data underlying the findings in their manuscript fully available?

Reviewer #1: Yes

Reviewer #2: Yes

Reviewer #3: Yes

5. Is the manuscript presented in an intelligible fashion and written in standard English?

Reviewer #1: Yes

Reviewer #2: Yes

Reviewer #3: Yes

6. Review Comments to the Author

Reviewer #1: Thank you very much for addressing each of the reviewers' comments. The authors have appropriately addressed the comments of the reviewers and I have no additional comments to add.

Reviewer #2: The authors are thanked for their thorough work revising the manuscript and have addressed reviewer comments in detail and modified text accordingly. No further comments!

Reviewer #3: The authors answered all my questions. I have no more comments for the authors, including concerns about research ethics.

7. PLOS authors have the option to publish the peer review history of their article (what does this mean?). If published, this will include your full peer review and any attached files.

Reviewer #1: No

Reviewer #2: No

Reviewer #3: No

---

## [Editor Report · Acceptance letter]

26 Jan 2024

PONE-D-23-28025R1 

PLOS ONE

Dear Dr. Sardesai, 

I'm pleased to inform you that your manuscript has been deemed suitable for publication in PLOS ONE. Congratulations! Your manuscript is now being handed over to our production team.

Kind regards, 

on behalf of

Dr. Wan-Jie Gu 

Academic Editor

PLOS ONE